# Drug-resistant TB prevalence study in 5 health institutions in Haiti

Jonathan Hoffmann[1]*, Carole Chedid[1,2], Oksana Ocheretina[3,4], Chloé Masetti[5], Patrice Joseph[4], Marie-Marcelle Mabou[4], Jean Edouard Mathon[4], Elie Maxime Francois[4], Juliane Gebelin[5], François-Xavier Babin[5], Laurent Raskine[5], Jean William Pape[4]

1 Fondation Mérieux, Direction Médicale et Scientifique, Lyon, France, 2 Département de Biologie, Ecole Normale Supérieure de Lyon, Lyon, France, 3 Division of Infectious Diseases, Department of Medicine, Center for Global Health, Weill Cornell Medical College, New York, New York, United States of America, 4 Les Centres GHESKIO, Port-au-Prince, Haiti, 5 Fondation Mérieux, Direction des Opérations Internationales, Lyon, France

☯ These authors contributed equally to this work.
* Jonathan.hoffmann@fondation-merieux.org

## Abstract

### Objectives

Tuberculosis (TB) is the leading infectious cause of death in the world. Multi-drug resistant TB (MDR-TB) is a major public health problem as treatment is long, costly, and associated to poor outcomes. Here, we report epidemiological data on the prevalence of drug-resistant TB in Haiti.

### Methods

This cross-sectional prevalence study was conducted in five health centers across Haiti. Adult, microbiologically confirmed pulmonary TB patients were included. Molecular genotyping (*rpoB* gene sequencing and spoligotyping) and phenotypic drug susceptibility testing were used to characterize rifampin-resistant MTB isolates detected by Xpert MTB/RIF.

### Results

Between April 2016 and February 2018, 2,777 patients were diagnosed with pulmonary TB by Xpert MTB/RIF screening and positive MTB cultures. A total of 74 (2.7%) patients were infected by a drug-resistant (DR-TB) *M. tuberculosis* strain. Overall HIV prevalence was 14.1%. Patients with HIV infection were at a significantly higher risk for infection with DR-TB strains compared to pan-susceptible strains (28.4% vs. 13.7%, adjusted odds ratio 2.6, 95% confidence interval 1.5–4.4, *P* = 0.001). Among the detected DR-TB strains, T1 (29.3%), LAM9 (13.3%), and H3 (10.7%) were the most frequent clades. In comparison with previous spoligotypes studies with data collected in 2000–2002 and in 2008–2009 on both sensitive and resistant strains of TB in Haiti, we observed a significant increase in the prevalence of the drug-resistant MTB Spoligo-International-Types (SIT) 137 (X2 clade: 8.1% vs. 0.3% in 2000–02 and 0.9% in 2008–09, p<0.001), 5 (T1 clade: 6.8% vs 1.9 in 2000–02 and 1.7% in

**Data Availability Statement:** The data underlying the results presented in the study have been uploaded with the manuscript as supplementary file (excel sheet "Study dataset").

**Funding:** This project was supported by the SPHaïtiLab project financed by the European Commission via DEVCO under the Supporting Public Health Institutes Programme and by the Fondation Mérieux. The funders had no role in study design, data collection and analysis, decision to publish, or preparation of the manuscript.

**Competing interests:** The authors have declared that no competing interests exist.

2008–09, $P = 0.034$) and 455 (T1 clade: 5.4% vs 1.6% and 1.1%, $P = 0.029$). Newly detected spoligotypes (SIT 6, 7, 373, 909 and 1624) were also recorded.

## Conclusion

This study describes the genotypic and phenotypic characteristics of DR-TB strains circulating in Haiti from April 2016 to February 2018. Newly detected MTB clades harboring multidrug resistance patterns among the Haitian population as well as the higher risk of MDR-TB infection in HIV-positive people highlights the epidemiological relevance of these surveillance data. The importance of detecting RIF-resistant patients, as proxy for MDR-TB in peripheral sites via molecular techniques, is particularly important to provide adequate patient case management, prevent the transmission of resistant strains in the community and to contribute to the surveillance of resistant strains.

## Introduction

Tuberculosis (TB) is a communicable disease caused by *Mycobacterium tuberculosis* (MTB). It is the leading infectious cause of death worldwide. In 2018, the death toll for TB among HIV-negative people was estimated to be 1.2 million, with an additional 251 000 deaths among HIV-positive people. The estimated global burden for the same year was about 10 million new cases [1]. Currently, about a quarter of the global population is latently infected, and at risk of developing active TB.

Although the Americas accounted only for 3% of the global TB cases in 2018, Haiti is one of the countries with the highest TB incidence in the Western hemisphere. In 2018, the country reported an incidence of 176 cases per 100,000 population, of which approximately 15% were cases of TB/HIV coinfection [2]. In Haiti, the mortality due to TB is estimated to be 9.2 per 100 000 population for HIV-negative patients, and 7.7 per 100 000 population for HIV-positive patients [3]. The overall HIV prevalence is 2% and has remained stable in the past years, with 160,000 people living with HIV in 2018 including about 8,400 to 10,000 adolescents for whom tailored interventions are needed to improve retention in care [4,5].

The definition of Multi Drug Resistance TB (MDR-TB) refers to strains that show resistance to at least both Rifampicin (RIF) and Isoniazid (INH) two of the primary four drugs used in TB treatment, whilst Drug Resistant TB (DR-TB) is may refer to strains that are resistant to one or more drugs used in TB treatment. ln Haiti, MDR-TB infections are a major public health issue as the treatment and the risk of poor outcomes are starkly increased [6,7]. The case management of such patients is costly for local healthcare systems and threatens to negatively impact of the progress made in the recent years in the fight against TB in Haiti [8]. In 2018 in Haiti, 94 patients were diagnosed with laboratory-confirmed MDR-TB or Rifampin Resistant TB (RR-TB) incidence of 5.1 per 100 000 population), and 91 were started on treatment [3]. There are two existing centers for the treatment of these patients in Haiti, and the capacity to perform Drug Susceptibility Testing (DST) to detect MDR-TB in Haiti remains limited outside the capital city of Port au Prince (Ouest department). However, since 2014, several peripheral laboratories have been equipped with GeneXpert® Systems (Cepheid, Sunnyvale, USA), enabling the molecular detection of rifampin resistant strains. Rifampin resistance serves as a proxy for MDR-TB diagnosis in low-resource settings. In December 2019, 27 public laboratories and subsidized health institutions across the country had access to GeneXpert machines.

In Haiti, the number of patients infected with Rifampin resistant MTB remains difficult to estimate [9]. Available studies on this topic mainly focus on the urban population of Port au Prince [6,10]. The present study was designed in collaboration with the National TB Program and the National Public Health Laboratory in 2014, and involved five health institutions located in the department of Ouest (GHESKIO INLR and GHESKIO IMIS), Nord (Hôpital Justinien du Cap Haïtien—HUJ), Sud (Hôpital Immaculée Conception des Cayes—HIC), and Centre (Hôpital Universitaire de Mirebalais—HUM). The general objective was to report the prevalence of DR-TB in these five study sites. For surveillance purposes, this study aimed to characterize the DR-TB strains circulating in Haiti from April 2016 to February 2018.

## Materials and methods

### Ethical statement

The study was approved by the Institutional Review Board of Weill Cornell Medical College (New York, USA), the Institutional Review Board of GHESKIO Centres (Port-au-Prince, Haiti) and the Haitian National Committee of Bioethics. Clinical and epidemiological data were extracted from patients' charts. As this was a retrospective clinical chart review, the requirement for informed consent was waived by the institutional review boards.

### Study sites

Five health institutions were selected to participate in the present study, based on the availability of a functional GeneXpert® System (Cepheid, Sunnyvale, USA) on site and considered to have large catchment areas for the population. The geographic distribution of the sites allowed to cover four out of 10 departments: Ouest (GHESKIO INLR and GHESKIO IMIS), Nord (Hôpital Justinien du Cap Haïtien—HUJ), Sud (Hôpital Immaculée Conception des Cayes— HIC) and Centre (Hôpital Universitaire de Mirebalais—HUM). Exclusion criteria included age under 15, negative smear microscopy, and signs of extrapulmonary TB.

### Tuberculosis screening and diagnostic algorithm

In the five participating health institutions, social workers administered a symptom checklist enquiring about chronic cough lasting $\geq$ 2 weeks, as well as other TB symptoms. Individuals who reported chronic cough were separated from other patients and referred for same-day physician evaluation. In the peripheral sites (HIC, HUJ, HUM), three sputum samples were collected per TB suspect: two were collected during the first consultation, then a sterile container was given to the patient for collection of the third specimen the next morning. Smear microscopy (Ziehl-Neelsen staining) was conducted. The third sputum specimen was refrigerated upon collection and transported to the biosafety level-3 Rodolphe Mérieux Reference Laboratory in the GHESKIO IMIS (Port-au-Prince). Xpert MTB/RIF testing was done directly on an early-morning specimen, in accordance with guidelines from the Haitian Ministry of Health [11] and the manufacturer's instructions. A single positive result from the smear examination led to the initiation of anti-TB treatment. HIV testing was conducted using rapid antibody tests (Determine; Alere, Waltham, MA, USA). The protocol conducted by the GHESKIO centers (INLR and IMIS) was slightly different as only two sputum specimens were collected per patient, and no microscopy testing was performed. Following internal diagnostics protocol, a digital chest radiograph (CXR) was performed on-site combined with Xpert MTB/RIF testing. All sputum specimens collected in the present study were cultured on liquid media (BACTEC MGIT 960, Becton Dickenson, Franklin Lakes, NJ, USA) and solid media (Lowenstein-Jensen).

## Phenotypic and genotypic drug-resistant TB assessment

DST to first- and second-line anti tuberculosis drugs was conducted for all samples with rifampin resistance detected by Xpert MTB/RIF. First-line DST was performed with BACTEC MGIT 960 SIRE and PZA kits as previously described [12]. Drug-resistant TB strains were further characterized using an in-house Luminex PCR-based spoligotyping assay and by *rpoB* gene sequencing as previously described [12,13]. Spoligotype International Type (SIT) patterns were assigned using the SITVIT2 database [14].

## Data analysis

Sociodemographic information, TB diagnosis and HIV status were collected using standardized clinical report forms. Data were cleaned and analyzed in R studio software (version 3.6.2). As the sample size was small, discrete variables were analyzed using Fisher's Exact test with Bonferroni's post-hoc [15]. Normal, continuous variables were analyzed using Student's t-test. Non-normal, continuous variables were analyzed using the Mann-Whitney or Kruskal-Wallis rank tests with Dunn's post-hoc [16] when necessary. For logistic regression, if missing data exceeded 10% of the sample size, the variable was not considered. Otherwise, missing data were replaced by the most frequent group (categorical variables) or the mean (continuous variables). Predictors were first evaluated in univariate logistic analyses, then models were adjusted for sociodemographic factors.

## Results

### Sociodemographic and clinical characteristics of the participants

From April 2016 to February 2018, 2,777 microbiologically confirmed pulmonary TB patients were enrolled in five study sites across Haiti. Initially, the aim of the study was to enroll 1,000 new TB patients and 250 re-treatment cases, with equal sampling from each site. However, country-wide strike movements hindered patient recruitment in peripheral study sites, prompting increased recruitment in central GHESKIO clinical centers (IMIS and INLR), hence the unequal sample size per site. All study sites were comparable regarding the age, sex ratio, and HIV prevalence of recruited participants (Table 1). The prevalence of HIV was 14.1% (391/2764). The frequency of retreatment was unequal between sites, with a lower frequency in the HUJ and HUM centers (center and northern sites) compared to the remaining

**Table 1. Sociodemographic and clinical characteristics of the cohort.**

|  | ALL | INLR | HIC | HUJ | HUM | IMIS | *P* |
|---|---|---|---|---|---|---|---|
|  | $N_T$ = 2777 | $N_T$ = 2049 | $N_T$ = 148 | $N_T$ = 50 | $N_T$ = 75 | $N_T$ = 455 |  |
| Age (years), median (IQR) | 31 (24–40) | 31 (25–41) | 28.5 (23–39) | 27 (23–40.75) | 31 (24.5–44.5) | 30 (23–39) | *0.132* |
| Sex (male), % (N) | 56.6% (1572/2777) | 58.1% (1190/2049) | 55.4% (82/148) | 60% (30/50) | 50.7% (38/75) | 51% (232/455) | *0.060* |
| HIV positivity, % (N) | 14.1% (391/2764) | 14.2% (291/2044) | 19.9% (29/146) | 12% (6/50) | 11.4% (8/70) | 12.6% (57/454) | *0.272* |
| Treatment category |  |  |  |  |  |  |  |
| New cases, % (N) | 86.5% (2401/2777) | 85.8% (1759/2049) | 84.5% (125/148) | 96% (48/50) | 96% (72/75) | 87.3% (397/455) | ***0.014*** |
| Relapse, % (N) | 8.5% (236/2777) | 8.5% (175/2049) | 13.5% (20/148) | 2% (1/50) | 4% (3/75) | 8.1% (37/455) | *0.058* |
| Treatment after interruption % (N) | 4% (110/2777) | 4.5% (92/2049) | 1.4% (2/148) | NA | NA | 3.5% (16/455) | ***0.049*** |
| Treatment after failure, % (N) | 1.1% (30/2777) | 1.1% (23/2049) | 0.7% (1/148) | 2% (1/50) | NA | 1.1% (5/455) | *0.823* |
| Drug resistance, % (N) | 2.7% (74/2758) | 1.7% (34/2045) | 0.7% (1/138) | 4.1% (2/49) | 4.2% (3/71) | 7.5% (34/455) | *<**0.001*** |

Data were analyzed using Fisher's Exact text or the Mann-Whitney U test (p-values given for all categories). Pairwise differences were assessed using Bonferroni or Dunn's post hoc. Stars indicate statistically different groups.

three sites. Among the 2,777 TB patients diagnosed with the Xpert MTB/RIF molecular test, 74 (2.7%) had TB isolates displayed a concordant genotypic/phenotypic DR-TB profiles. The prevalence of detected DR-TB strains was higher in te IMIS center compared to the rest of the partner sites.

### Phenotypic diversity of detected drug-resistant TB strains

Among the 74 drug-resistant TB strains identified by DST, 12 strains (16.2%) displayed a mono-resistance to at least 1 antibiotic (drug-resistant TB strain; DR-TB) and 62 strains (83.8%) displayed a multi-drug resistant (MDR-TB) profile (defined by resistance to at least rifampin and isoniazid). Resistance phenotypes to INH and RIF were predominant (87.8% and 91.9%, respectively).

Details of monoresistance and multi-drug resistant profiles are shown in S1 Table and S1 Fig. Sixty-two (83.8%) DR-TB strains were identified in new TB cases, 11 (14.9%) in relapse TB cases, and 1 (1.4%) in subjects under retreatment after failure (Table 2).

### Comparison of clinical and sociodemographic factors between drug-susceptible and drug-resistant TB patients

We then compared the sociodemographic and clinical characteristics of DR-TB and DS-TB patients. Patient characteristics were comparable (age group, sex ratio, and treatment category), but HIV prevalence was significantly higher in DR-TB patients (28.4% vs. 13.7%, p = 0.002) (Table 2). Logistic regression analyses confirmed that HIV-positive patients recruited in this cohort were at higher risk for detection of DR-TB strains. After adjusting for age, sex, and study site of recruitment, HIV-positive patients were 2.5 times more likely to be diagnosed with DR-TB than HIV-negative patients.

### Diversity of circulating drug-resistant MTB strains

The repartition of drug-resistant *M. tuberculosis* strains in Haiti during this study period was then assessed (Fig 1). T1, LAM9, LAM1 and H3 were the most frequently detected clades

**Table 2. Comparison of drug-susceptible and resistant TB patients and evaluation of associated risk factors.**

| | Total | Drug sensitive | Drug resistant | | Logistic regression—association with drug resistance | | | | | | | | |
| | | | | | Univariate analysis | | | | Multivariate analysis* | | | | |
| | $N_T = 2777$ | $N_T = 2684$ | $N_T = 74$ | P | OR | 2.50% | 97.50% | P | aOR | lower | upper | P | AIC |
| Sex (male) | 56.6% (1572/2777) | 56.9% (1526/2684) | 51.4% (38/74) | 0.345 | 0.855 | 0.616 | 1.188 | 0.347 | - | - | - | - | - |
| HIV positivity | 14.1% (391/2764) | 13.7% (365/2672) | 28.4% (21/74) | **0.002** | 2.504 | 1.463 | 4.139 | **0.001** | 2.585 | 1.478 | 4.383 | **0.001** | 643.28 |
| Age (years; median, IQR) | 31 (24–40) | 30.5 (24–40) | 33 (24–43.75) | 0.313 | 1.012 | 0.995 | 1.029 | 0.162 | - | - | - | - | - |
| Treatment category | | | | | | | | | | | | | |
| New cases | 86.5% (2401/2777) | 86.7% (2328/2684) | 83.8% (62/74) | 0.486 | 0.79 | 0.437 | 1.553 | 0.462 | 0.756 | 0.152 | 13.737 | 0.788 | 650.37 |
| Relapse | 8.5% (236/2777) | 8.2% (219/2684) | 14.9% (11/74) | 0.052 | 1.965 | 0.968 | 3.636 | 0.043 | 1.397 | 0.247 | 26.408 | 0.756 | 650.37 |
| Treatment after interruption | 4% (110/2777) | 4.1% (109/2684) | 0 | 0.118 | 0 | 0 | 5789.26 | 0.981 | - | - | - | - | - |
| Treatment after failure | 1.1% (30/2777) | 1% (28/2684) | 1.4% (1/74) | 0.547 | 1.299 | 0.072 | 6.216 | 0.798 | 0 | 0 | 2529.87 | 0.98 | 650.37 |

Data were compared with Fisher's Exact Test. OR: odds ratio. aOR: Adjusted odds ratio. IQR: Interquartile range. AIC: Akaike Information Criterion.

*Models were adjusted for age, sex, and study site. Fifteen out of 2,777 smear-positive samples were tested GeneXpert MTB/RIF negative (MTB not detected) and 4 isolates could not be cultured for DST.

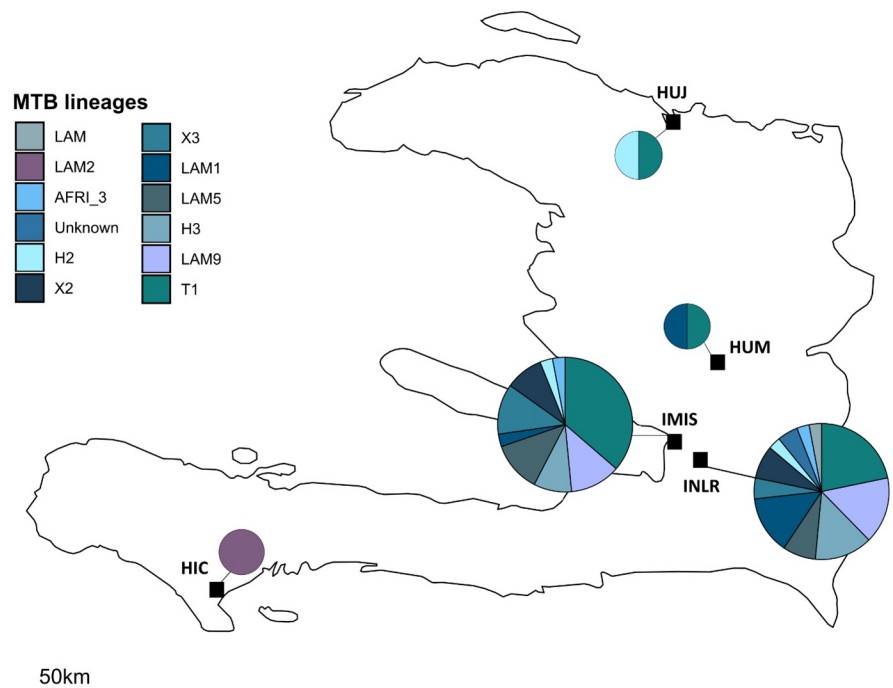

**Fig 1. Repartition of drug-resistant M. tuberculosis clades in Haiti (n = 74).** Geographic repartition of DR-TB clades in partner study sites. Data are given for all patients with known spoligotypes. Pie chart size correlates with sample size in each site (Gheskio: n = 34. IMIS: n = 34. HUM: n = 3. HUJ: n = 2. HIC: n = 1).

among all DR-TB patients. In the two sites with the most recruited patients (GHESKIO IMIS and INLR), T1 was also the most frequently detected clade, however, a greater diversity of clades was observed in the INLR center. Frequencies of detected DR-TB clades and spoligotypes were compared with data collected in 2002 and in 2009 [13] (Table 3). The prevalence of X2 strains (SIT 137) significantly increased from 0.3% and 0.9% in 2002 and 2009, respectively, to 8.1% (6/74) in our study (P <0.001). Three of these strains had the same resistance phenotype (STR+INH+RIF+EMB+ETH), the same *rpoB* mutation (S531L), and were detected in the same study site (GHESKIO INLR) (Table 4). A similar phenomenon was observed with T1 strains (SIT 5), carrying the D516V *rpoB* mutation, with the same resistance profiles (STR +INH+RIF+EMB), and detected in the same site (GHESKIO INLR). Moreover, we recorded rare occurrences of MDR-TB strains that were not detected in these previous studies (T1 SIT 373 and 7, LAM SIT 1624, EAI-SOM SIT 6 and SIT 909 with unknown clade) among new TB cases (S2 Table). Finally, some of the strains that were highly prevalent in previous works (LAM2) were less frequent in our study (1.4% vs. 5.8% in 2002 and 4.4% in 2009).

## Relationship between *rpoB* mutation and lineage

*rpoB* sequencing was performed on DR-TB isolates (S2 Fig). S531L was the most frequently detected mutation (44/74, 59.4%) followed by S531W and D516V (7/74, 9.4% respectively). *rpoB* mutations were heterogeneously distributed within the detected clades (S3 Fig). As expected, the highest mutation diversity was observed in the most frequent lineage (T1), and the most frequent *rpoB* mutation (S531L) was detected in every lineage. Interestingly, S531L mutations were detected in at least 50% of isolates for every lineage except LAM9, in which

**Table 3. Spoligotyping data of TB isolates in Haiti.**

| SIT | Clade | TB isolates N (%) | | | |
| --- | --- | --- | --- | --- | --- |
| | | 2000–2002* | 2008–2009* | 2016–2018 | |
| | | $N_T = 379$ | $N_T = 758$ | $N_T = 74$** | *P* |
| 42 | LAM9 | 27 (7.1) | 54 (7.1) | 10 (13.5) | 0.149 |
| 53 | T1 | 27 (7.1) | 46 (6.1) | 7 (9.5) | 0.433 |
| 93 | LAM5 | 15 (4) | 33 (4.4) | 7 (9.5) | 0.135 |
| 20 | LAM1 | 26 (6.9) | 42 (5.5) | 4 (5.4) | 0.693 |
| 91 | X3 | 29 (7.7) | 31 (4.1) | 5 (6.8) | 0.031 |
| 137 | X2 | 1 (0.3) | 7 (0.9) | 6 (8.1) | <**0.001** |
| 5 | T1 | 7 (1.9) | 13 (1.7) | 5 (6.8) | **0.034** |
| 50 | H3 | 37 (9.8) | 79 (10.4) | 2 (2.7) | 0.084 |
| 455 | T1 | 6 (1.6) | 8 (1.1) | 4 (5.4) | **0.029** |
| 2 | H2 | 37 (9.8) | 72 (9.5) | 3 (4.1) | 0.287 |
| 77 | T1 | 4 (1.1) | 11 (1.5) | 3 (4.1) | 0.149 |
| 51 | T1 | 17 (4.5) | 26 (3.4) | 2 (2.7) | 0.671 |
| 294 | H3 | 1 (0.3) | 2 (0.3) | 2 (2.7) | **0.039** |
| 408 | AFRI_3 | - (-) | 4 (0.5) | 1 (1.4) | **0.019** |
| 17 | LAM2 | 22 (5.8) | 33 (4.4) | 1 (1.4) | 0.218 |
| 373 | T1 | - (-) | - (-) | 1 (1.4) | 1.000 |
| 578 | LAM1 | 3 (0.8) | 7 (0.9) | 1 (1.4) | 0.761 |
| 714 | H3 | 4 (1.1) | 3 (0.4) | 1 (1.4) | 0.189 |
| 909 | No clade | - (-) | - (-) | 1 (1.4) | 1.000 |
| 1624 | LAM | - (-) | - (-) | 1 (1.4) | 1.000 |
| 6 | EAI-SOM | - (-) | - (-) | 1 (1.4) | 0.061 |
| 7 | T1 | - (-) | - (-) | 2 (2.7) | **0.003** |

*Data from Ocheretina, O. *et al*. Journal of Clinical Microbiology 51, 2232–2237 (2013). Both studies include DS and DR-TB strains.

**4 samples failed to generate an interpretable pattern (SIT). SIT and Clades were retrieved from the SITVIT2 international database. Data were compared with Fisher's Exact Test.

S531W was the most frequent mutation. No difference was observed when stratifying data according to HIV status (data not shown).

## Discussion

In this study, we reported the occurrence of DR-TB infections in patient cohorts across five study sites in Haiti. We evaluated their distribution according to selected sociodemographic parameters, and we described the phenotypic and genotypic characteristics of isolated DR-TB strains.

First, we observed that patients living with HIV infection were at a significantly higher risk for infection with DR-TB strains compared to pan-susceptible strains (31% vs. 15%, aOR 2.5, 95%CI 1.5–4.1, p < 0.001). Comparatively, in Haiti, reported rates of HIV infection are of 16% in TB patients regardless of the drug-susceptibility status, and 2% in the general population [17]. Increased epidemiological interactions between HIV and MDR-TB infections compared to pan-susceptible TB infections have been described since the late nineties [18]. Data on this topic in Haiti are scarce, but a 2006 study reported significantly higher rates of MDR-TB in HIV-positive than in HIV-negative patients [7], which is corroborated by our results. The reason for the interactions between HIV and MDR-TB remains unclear: explanatory hypotheses available in the literature include increased anti-TB drug malabsorption in HIV-positive

**Table 4. Description of DR-TB cases and characterization of genotypic and phenotypic drug-resistances by SIT.**

| SIT | CLADE | SITE | PID | AGE | SEX | TREATMENT CATEGORY | HIV status | *rpoB* mutation | Drug Susceptibility Testing | | | | | |
|-----|-------|------|-----|-----|-----|--------------------|------------|----------------|-----|-----|-----|-----|-----|-----|
| | | | | | | | | | STR | INH | RIF | EMB | PZA | ETH |
| 137 | X2 | GHESKIO | GH-0128 | 62 | M | N | P | S531L | **R** | **R** | **R** | **R** | **S** | **R** |
| | | GHESKIO | GH-0477 | 25 | M | N | N | S531L | **R** | **R** | **R** | **R** | **S** | **R** |
| | | GHESKIO | GH-0573 | 68 | F | N | N | S531L | **R** | **R** | **R** | **R** | **S** | **R** |
| | | IMIS | IMIS-040 | 71 | M | N | N | S531L | S | R | R | S | R | R |
| | | IMIS | IMIS-388 | 57 | F | N | N | S531L | S | R | R | R | R | R |
| | | IMIS | IMIS-420 | 19 | M | R | I | S531L | S | R | R | S | R | S |
| 5 | T1 | GHESKIO | GH-0235 | 17 | M | N | P | D516V | **R** | **R** | **R** | **R** | **S** | **S** |
| | | GHESKIO | GH-0269 | 29 | M | N | N | D516V | **R** | **R** | **R** | **R** | **S** | **S** |
| | | GHESKIO | GH-1017 | 21 | F | N | N | D516V | **R** | **R** | **R** | **R** | **S** | **S** |
| | | GHESKIO | GH-1980 | 45 | M | N | N | D516V | **R** | **R** | **R** | **R** | **S** | **S** |
| | | IMIS | IMIS-239 | 69 | F | R | P | S531L | S | S | R | S | S | S |
| 455 | T1 | GHESKIO | GH-0640 | 26 | M | R | N | S531L | **S** | **S** | **R** | **S** | **S** | **S** |
| | | IMIS | IMIS-344 | 29 | M | N | P | S531L | **S** | **S** | **R** | **S** | **S** | **S** |
| | | IMIS | IMIS-210 | 18 | M | N | N | S531L | R | R | R | R | R | S |
| | | HUM | PIH-005 | 30 | M | R | N | S531L | R | R | R | R | S | R |

SIT and clades were retrieved from the SITVIT2 international database. Sex: M = male, F = female; Treatment category: N = new case, R = Relapse; HIV status: P = positive; N = Negative; Drug susceptibility testing: R = Resistant; S = Susceptible; Drugs: STR = streptomycin, INH = isoniazid, RIF = Rifampin, EMB = ethambutol, PZA = Pyrazinamide), ETH = ethionamide. The shaded areas represent identical genotypic and phenotypic resistance profiles for the same SIT.

patients, or enhanced fitness of MDR-TB strains compared to pan-susceptible strains in HIV-positive hosts [19]. However, current evidence points to an increase in the risk of primary MDR-TB infection in HIV-positive patients, rather than acquired drug-resistance [20].

Secondly, we studied the phenotypic and genotypic diversity of the DR-TB strains isolated from our cohort. Most strains with RIF resistance upon GeneXpert testing also displayed drug or multi-drug resistant phenotypes upon DST. The most frequent resistance phenotypes detected were resistance to Isoniazid and Rifampin, and resistance to all four first line anti-TB drugs. Sequencing analyses identified two high frequency *rpoB* mutations (S531L and S531W) and 12 low-frequency *rpoB* mutations on different loci. Five strains with discrepant phenotypic and genotypic results for resistance to RIF were isolated. Strains exhibiting these discrepancies have been described in previous works in Haiti as possibly harboring subcritical levels of resistance to RIF [12,21]. In our cohort, sequencing analyses identified either T508A or silent T508T *rpoB* mutations in these five discrepant strain isolates, and their SIT numbers were 20 and 50 respectively, which is consistent with earlier findings [12]. In addition, strains harboring a L511P *rpoB* mutation have been detected in our cohort; while this has been previously found in phenotypically RIF susceptible strains [12], they were phenotypically drug-resistant in our case.

Thirdly, we aimed to identify the MTB clades detected in the cohort's DR-TB patients. Indeed, molecular epidemiology is now an important tool to determine MTB transmission patterns, as it complements classic epidemiologic contact tracing and allows investigators to better characterize transmission dynamics. In this study, T1, LAM9, LAM1 and H3 were the most frequently detected clades among all DR-TB patients. In the two sites with the most recruited patients (GHESKIO IMIS and INLR), T1 was the most frequently detected clade as well, but increased diversity was observed in the INLR center. These observations are consistent with a review of MTB drug-resistance and associated genotypic clades observed in 3 French Departments of the Caribbean (Guadeloupe, Martinique and French Guiana) over a seventeen-year period (January 1995–December 2011) [22]. This review reports that T, LAM,

and H were the most common clades, respectively accounting for 29.9% (358/1199), 23.9% (286/1199), and 22.1% (265/1199) of all DR-TB isolates. Moreover, a previous spoligotyping study of 758 TB strains collected in patients presenting to the 6 largest TB centers in and around Port-au-Prince (2008-to-2009 MDR-TB survey [6]) revealed that H3 (10.4%, SIT 50), H2 (9.5%, SIT 2), LAM9 (7.1%, SIT42), T1 (6.1%, SIT 53) and LAM-1 (5.5%, SIT 20) were the 5 most prevalent clades circulating in Haiti during this period [13]. Other spoligotyping data from a project conducted by GHESKIO and Pasteur Institute Guadeloupe from 2000 to 2002 showed that H3 (9.8%, SIT 50), H2 (9.8%, SIT 2), LAM9 (7.1%, SIT42), T1 (7.1%, SIT 53) and LAM-1 (6.9%, SIT 20) were also the 5 most prevalent TB clades detected in 378 GHESKIO patients [13]. Here, a comparative analysis of DR-TB strains spoligotypes circulating in Haiti revealed a resurgence in the number of cases caused by strains with a previously low prevalence (i.e. SIT 137, 5, 455 and 294) or even unknown (SIT 373, 6, 7 909 and 1624). Phenotypic and genotypic analysis of individual MTB strains revealed several identical patterns of MDR-TB. These observations suggest that patients with genetically identical strains may have been infected by a common index case or may have been part of a larger cluster of cases.

The Luminex spoligotyping method currently used by GHESKIO in combination with the GeneXpert MTB/RIF was adapted as a first-line, high-throughput tool for MTB genotyping in resource-limited countries. Luminex spoligotyping allows real-time typing and can be used for multiple clinical and public health purposes in Haiti, such as epidemiological investigation through community-based active case finding (ACF), contact tracing, and MTB strain characterization during clinical trials of pulmonary MDR-TB treatments [23–27]. Data from a 2014–2015 retrospective cohort analysis using the GHESKIO ACF campaign revealed that the prevalence of TB and HIV in slums of Port-au-Prince was respectively four and five times higher than national estimates [23]. Active case finding for TB and HIV should be expanded to other slum populations in Haiti as part of routine programmatic activities to increase the detection rate of TB cases.

## Conclusion

Overall, our study showed that people living with HIV in Haiti were particularly at risk for drug-resistant TB, which is a major public health issue on the island. The identified MTB clades were consistent with similar works conducted in the Caribbean, and several MTB clades harboring drug resistance patterns were either newly identified or increasingly detected among the Haitian population. These observations demonstrate that MTB strain genotyping, identification, and surveillance of specific *M. tuberculosis* SITs are essential to better understand the dynamics of DR-TB strain transmission, and to design adapted TB control measures in Haiti. The use of Xpert MTB/RIF testing increased the detection rate of patients with bacteriologically-confirmed TB, and an additional surrogate spoligotyping method is useful to identify MDR-TB clades among newly diagnosed TB cases, to differentiate reactivation from re-infection, to discover more virulent strains, and to monitor the spread of new types.

## Supporting information

**S1 Fig. Description of DR-TB strains phenotypes among all patients with available DST results.** Total of 74 DR-TB strains including 12 mono-drug resistant strains and 62 multi-drug resistant TB strains. GHESKIO (INLR): n = 34. IMIS: n = 34. HUM: n = 3. HUJ: n = 2. HIC: n = 1. I: Isoniazid. R: Rifampin. S: Streptomycin. E: Ethambutol. PZA: Pyrazinamide. KM: Kanamycin.
(DOCX)

**S2 Fig. *rpoB* genotypic diversity of DR-TB isolates (n = 74).**
(DOCX)

**S3 Fig. Frequency of detected rpoB mutation in each identified drug-resistant *M. tuberculosis* lineage (n = 74).** Data are given for all patients with known spoligotypes.
(DOCX)

**S1 Table. Summary of resistance profiles of DR-TB isolates identified in new TB cases, relapse, treatment after failure or treatment after interruption.**
(DOCX)

**S2 Table. Mono-resistance profile of DR-TB isolates identified in new TB cases, relapse, treatment after failure, treatment after interruption.**
(DOCX)

**S3 Table. Multi-resistant profile of DR-TB isolates identified in new TB cases, relapse, treatment after failure, treatment after interruption.**
(DOCX)

**S4 Table. Summary of resistance profiles of DR-TB isolates identified in new TB cases, relapse, treatment after failure, treatment after interruption.**
(DOCX)

**S5 Table. Distribution of DR-TB lineages (SIT) among new TB cases, relapse and treatment after failure.**
(DOCX)

**S1 Dataset.**
(XLSX)

## Acknowledgments

We are very grateful to the National Laboratory for Public Health (LNSP) and the National Programme for TB (PNLT) for their collaboration in the design of the study, and to the administrative and laboratory staff in the five participating sites for sample collection and processing.

## Author Contributions

**Conceptualization:** Oksana Ocheretina, Chloé Masetti, Patrice Joseph, Marie-Marcelle Mabou, Jean Edouard Mathon, Elie Maxime Francois, Juliane Gebelin, François-Xavier Babin, Jean William Pape.

**Data curation:** Jonathan Hoffmann, Carole Chedid, Oksana Ocheretina, Patrice Joseph, Marie-Marcelle Mabou, Jean Edouard Mathon, Elie Maxime Francois, Laurent Raskine, Jean William Pape.

**Formal analysis:** Oksana Ocheretina.

**Funding acquisition:** Chloé Masetti, Juliane Gebelin, François-Xavier Babin, Jean William Pape.

**Investigation:** Jonathan Hoffmann, Carole Chedid, Oksana Ocheretina, Patrice Joseph, Marie-Marcelle Mabou, Jean Edouard Mathon, Elie Maxime Francois, Jean William Pape.

**Methodology:** Oksana Ocheretina, Patrice Joseph, Marie-Marcelle Mabou, Jean Edouard Mathon, Elie Maxime Francois, Jean William Pape.

**Project administration:** Chloé Masetti, François-Xavier Babin, Jean William Pape.

**Software:** Jonathan Hoffmann, Carole Chedid, Laurent Raskine.

**Supervision:** Oksana Ocheretina, Chloé Masetti, Juliane Gebelin, François-Xavier Babin, Jean William Pape.

**Validation:** Oksana Ocheretina, Patrice Joseph, Marie-Marcelle Mabou, Jean Edouard Mathon, Elie Maxime Francois, Juliane Gebelin, Laurent Raskine, Jean William Pape.

**Visualization:** Jonathan Hoffmann, Carole Chedid.

**Writing – original draft:** Jonathan Hoffmann, Carole Chedid, Chloé Masetti, Laurent Raskine.

**Writing – review & editing:** Jonathan Hoffmann, Carole Chedid, Oksana Ocheretina, Chloé Masetti, Patrice Joseph, Marie-Marcelle Mabou, Jean Edouard Mathon, Elie Maxime Francois, Juliane Gebelin, François-Xavier Babin, Laurent Raskine, Jean William Pape.

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
