## [Decision Letter · Decision Letter 0]

21 Dec 2020

PONE-D-20-31826

Drug-resistant TB prevalence study in 5 health institutions in Haiti

PLOS ONE

Dear Dr. HOFFMANN,

Thank you for submitting your manuscript to PLOS ONE. After careful consideration, we feel that it has merit but does not fully meet PLOS ONE’s publication criteria as it currently stands. Therefore, we invite you to submit a revised version of the manuscript that addresses the points raised during the review process

We look forward to receiving your revised manuscript.

Kind regards,

Shampa Anupurba, MD

Academic Editor

PLOS ONE

Journal Requirements:

2. Please note that all PLOS journals ask authors to adhere to our policies for sharing of data and materials: https://journals.plos.org/plosone/s/data-availability. According to PLOS ONE’s Data Availability policy, we require that the minimal dataset underlying results reported in the submission must be made immediately and freely available at the time of publication. As such, please remove any instances of 'unpublished data' or 'data not shown' in your manuscript and replace these with either the relevant data (in the form of additional figures, tables or descriptive text, as appropriate), a citation to where the data can be found, or remove altogether any statements supported by data not presented in the manuscript.

4. Please remove your figures from within your manuscript file, leaving only the individual TIFF/EPS image files, uploaded separately.  These will be automatically included in the reviewers’ PDF.

5.We note that [Figure(s) 1] in your submission contain map images which may be copyrighted. All PLOS content is published under the Creative Commons Attribution License (CC BY 4.0), which means that the manuscript, images, and Supporting Information files will be freely available online, and any third party is permitted to access, download, copy, distribute, and use these materials in any way, even commercially, with proper attribution. For these reasons, we cannot publish previously copyrighted maps or satellite images created using proprietary data, such as Google software (Google Maps, Street View, and Earth). For more information, see our copyright guidelines: http://journals.plos.org/plosone/s/licenses-and-copyright.

a.    You may seek permission from the original copyright holder of Figure(s) [1] to publish the content specifically under the CC BY 4.0 license. 

Additional Editor Comments (if provided):

Line 58-delete 'is' in 'is may refer to... '

Line 61- 'threatens to negatively impact of....' may be rewritten as 'threatens to have negative impact on the progress....'

Line 63- 'TB (RR-TB) incidence...' may be rewritten as 'TB (RR-TB) with an incidence...'

Line 116- 'DST to...' may be rewritten as 'DST of...'

Line 213- delete 'of' in '....reported rates of HIV infection are of 16%...'

Reviewers' comments:

Reviewer's Responses to Questions

**Comments to the Author**

1. Is the manuscript technically sound, and do the data support the conclusions?

Reviewer #1: Yes

2. Has the statistical analysis been performed appropriately and rigorously? 

Reviewer #1: Yes

3. Have the authors made all data underlying the findings in their manuscript fully available?

Reviewer #1: Yes

4. Is the manuscript presented in an intelligible fashion and written in standard English?

Reviewer #1: Yes

5. Review Comments to the Author

Reviewer #1: "Drug-resistant TB prevalence study in 5 health institutions in Haiti"

I congratulate the authors who have brought data on DR-TB in Haiti pertaining to patients and mycobacterial characteristics under difficult circumstances of strike.

From this manuscript it is evident that the authors wanted to conduct this study in five sites across Haiti by enrolling 1000 new and 250 re-treatment TB cases and to characterise the DR-TB isolates further. However, due to strike in the peripheral regions, the study was mainly conducted in the central region, IMIS and INLR (contributing 2504 enrolments) as compared to peripheral regions (273 enrolments) from April 2016 to February 2018. Out of these, 2401 were new cases while 376 were re-treatment cases. I wonder if the samples/patients from the peripheral sites (HIC, HUJ and HUM) were referred to central laboratories for diagnosis to make the title relevant as DR-TB prevalence in regions catered by 5 health institutions.

Minor Comments:

Abstract:

1. Objectives: Please write the objectives of the study clearly.

2. Results: The authors may give their own results only, avoiding comparison with the previous studies that may be done in the discussion part of the main manuscript.

3. Conclusions: The conclusions can be crisp and corroborate the objectives.

Manuscript:

1. Introduction: The part “including about 8400 to 10000 adolescents ------retention in care” may be avoided.

2. Line 59, MDR-TB infections may replaced with MDR-TB disease

3. The case management of such patients is costly for local healthcare systems and threatens to negatively impact of the progress made in the recent years in the fight against TB in Haiti. “of” may be deleted.

4. Line 94 to 95: Exclusion criteria included age under 15, negative smear microscopy, and signs of extrapulmonary TB. This sentence does not match with lines 109 to 110 where it is mentioned “The protocol conducted by the GHESKIO centers (INLR and IMIS) was slightly different as only two sputum specimens were collected per patient, and no microscopy testing was performed”. These two sites were main contributors for the study.

5. Line 136-137: How was sample size of 1250 arrived at? And why did the authors exceed the pre-determined sample size? Do 2777 cases represent all the cases during the study duration that satisfy the inclusion criteria or some other sampling techniques was used considering the strike in the regions.

6. Line 141: unequal between sites may be replaced with unequal among sites.

7. Line 154-156: Please clarify if the study had 62 isolates that were MDR among total 74 DR TB cases and also 62 DR TB isolates that were identified in new TB cases. Please also clarify the type of resistance found in new and re-treatment cases.

8. Table 2: In treatment after interruption, how was p value of 0.118 obtained when there was no DR-TB case?

9. Line 175-184: This part may be shifted to discussion.

10. While you use the word “cohort” in discussion in lines 225, 231 and 237, what do we understand? Can were replace it with “the tested isolates”

11. Line 269, can we make it “detection rate of HIV-TB co-infection”?

12. Conclusions: Please make the conclusions in-line with the objectives.

13. References: Many of the references are in French.

14. Supplementary material Table 1S: In the heading, Treatment column is depicted twice, please make it as Treatment after interruption and Treatment after failure.

15. Table 1S: Resistant to 3AB* - the line total should be 13.

16. Figure 1S: If you give number of isolates also, it will clarify the confusion of MDR-TB cases among total DR-TB cases.

6. PLOS authors have the option to publish the peer review history of their article (what does this mean?). If published, this will include your full peer review and any attached files.

Reviewer #1: **Yes: **Rahul Narang

---

## [Author Response · Author response to Decision Letter 0]

5 Feb 2021

Lyon, February 02nd 2021,

Re: “Drug-resistant TB prevalence study in 5 health institutions in Haiti” 

PONE-D-20-31826

Dear Dr. Shampa Anupurba,

We thank you and the reviewers for your constructive comments regarding this manuscript. We found that the Reviewer’s comments were helpful and we have addressed each suggestion to enhance the manuscript overall quality. As recommended, we have resubmitted a revised manuscript and justified our responses to each of the comments below (in blue) for your attention. 

“I congratulate the authors who have brought data on DR-TB in Haiti pertaining to patients and mycobacterial characteristics under difficult circumstances of strike. 

From this manuscript it is evident that the authors wanted to conduct this study in five sites across Haiti by enrolling 1000 new and 250 re-treatment TB cases and to characterise the DR-TB isolates further. However, due to strike in the peripheral regions, the study was mainly conducted in the central region, IMIS and INLR (contributing 2504 enrolments) as compared to peripheral regions (273 enrolments) from April 2016 to February 2018. Out of these, 2401 were new cases while 376 were re-treatment cases. I wonder if the samples/patients from the peripheral sites (HIC, HUJ and HUM) were referred to central laboratories for diagnosis to make the title relevant as DR-TB prevalence in regions catered by 5 health institutions. 

Minor Comments:

Abstract:

1. Objectives: Please write the objectives of the study clearly.

We clarified the objectives of this study which were : “The main objectives are to report epidemiology of TB in Haiti from April 2016 to February 2018 and depict molecular genotyping and phenotyping of drug-resistant TB isolates.”

2. Results: The authors may give their own results only, avoiding comparison with the previous studies that may be done in the discussion part of the main manuscript. 

This is indeed a good point, we have moved the following paragraph into the discussion: “In comparison with previous spoligotypes studies with data collected in 2000-2002 and in 2008-2009 on both sensitive and resistant strains of TB in Haiti, we observed a significant increase in the prevalence of the drug-resistant MTB Spoligo-International-Types (SIT) 137 ( X2 clade : 8.1% vs. 0.3% in 2000-02 and 0.9% in 2008-09, p<0.001), 5 (T1 clade: 6.8% vs 1.9 in 2000-02 and 1.7% in 2008-09, P=0.034) and 455 (T1 clade: 5.4% vs 1.6% and 1.1%, P=0.029). Newly detected spoligotypes (SIT 6, 7, 373, 909 and 1624) were also recorded.”

3. Conclusions: The conclusions can be crisp and corroborate the objectives. 

We have reformulated the conclusions of this abstract as follows: “Overall, this epidemiological surveillance study reveals an active circulation of drug-resistant TB isolates in the Haitian population, of new or well-known genotypes. In this study, the risk of infection with DR-TB isolates is higher in HIV-positive individuals. This study underscores the importance of routine screening and genotyping of tuberculosis isolates at peripheral sites using molecular techniques to provide adequate patient care, prevent transmission of resistant strains in the community, and contribute to the surveillance of resistant strains”

Manuscript:

4. Introduction: The part “including about 8400 to 10000 adolescents ------retention in care” may be avoided.

We agree, we have withdrawn the following statement: “including about 8,400 to 10,000 adolescents for whom tailored interventions are needed to improve retention in care(4,5).”

We have modified the sentence as follows: “The overall HIV prevalence is 2% and has remained stable in the past years, with 160,000 people living with HIV in 2018 (4).”

5. Line 59, MDR-TB infections may replaced with MDR-TB disease

We also agree, we have made the following change: “ln Haiti, MDR-TB diseases are a major public health issue as the treatment and the risk of poor outcomes are starkly increased(6,7).”

6. The case management of such patients is costly for local healthcare systems and threatens to negatively impact of the progress made in the recent years in the fight against TB in Haiti. “of” may be deleted. 

Changes: “The case management of such patients is costly for local healthcare systems and threatens to negatively impact the progress made in the recent years in the fight against TB in Haiti(8).”

7. Line 94 to 95: Exclusion criteria included age under 15, negative smear microscopy, and signs of extrapulmonary TB. This sentence does not match with lines 109 to 110 where it is mentioned “The protocol conducted by the GHESKIO centers (INLR and IMIS) was slightly different as only two sputum specimens were collected per patient, and no microscopy testing was performed”. These two sites were main contributors for the study.

You are right, this is an error on our part, the exclusion criteria were as follows (modification line 94): “Exclusion criteria included age under 15, negative Xpert MTB/RIF, and signs of extrapulmonary TB”.

8. Line 136-137: How was sample size of 1250 arrived at? And why did the authors exceed the pre-determined sample size? Do 2777 cases represent all the cases during the study duration that satisfy the inclusion criteria or some other sampling techniques was used considering the strike in the regions.

We have added an explanatory paragraph about this in the Methods (lines 110-119). 

Because of the strike, we suspected there would be a lesser geographic diversity among included patients and Mtb strains, which led us to increase recruitment in the central GHESKIO clinical centers (IMIS and INLR) in an attempt to be more exhaustive. 

The sampling techniques were the same as planned, so the 2777 cases do represent those cases which satisfy the inclusion criteria.

9. Line 141: unequal between sites may be replaced with unequal among sites.

We have corrected this sentence as follow : “The frequency of retreatment was unequal among sites, with a lower frequency in the HUJ and HUM centers (center and northern sites) compared to the remaining three sites."

10. Line 154-156: Please clarify if the study had 62 isolates that were MDR among total 74 DR TB cases and also 62 DR TB isolates that were identified in new TB cases. Please also clarify the type of resistance found in new and re-treatment cases.

Indeed, this is a point that we have clarified. Briefly, a total of 74 DR-TB strains were identified in this study, of which 12 (16.2%) were characterized as having mono-phenotypic resistance to drugs and 62 (83.8%) with a multi-drug resistant profile. Of the total 74 DR-TB, 62 strains were identified in new cases, 11 in relapses and 1 in a person under retreatment following a treatment failure. Among the 62 DR-TB strains identified in new cases: 54 are considered MDR-TB and 8 as mono-resistant. 

We have added 3 tables in the supplementary data: table 2S, 3S and 4S: 

- Table 2S: Mono-resistance profile of DR-TB isolates identified in different groups of people (new cases, relapse, treatment after failure, treatment after interruption);

- Table 3S: Multi-resistant profile of DR-TB isolates identified in different groups of people (new cases, relapse, treatment after failure, treatment after interruption)

- Table 4S: Summary of resistance profiles of DR-TB isolates identified in different groups of people (new cases, relapse, treatment after failure, treatment after interruption).

Also, clarification has been brought to the legend of Figure 1S: “Total of 74 DR-TB strains including 12 mono-drug resistant strains and 62 multi-drug resistant TB strains”.

11. Table 2: In treatment after interruption, how was p value of 0.118 obtained when there was no DR-TB case?

This is indeed aberrant and has been withdrawn. 

12. Line 175-184: This part may be shifted to discussion.

Agreed. We have rephrased this paragraph and shifted part of it to the Discussion (lines 286-290) to avoid citing article tables in the Discussion.

13. While you use the word “cohort” in discussion in lines 225, 231 and 237, what do we understand? Can were replace it with “the tested isolates”

You are right, we have modified sentences as follow:

- Line 226: “In this study, we reported the occurrence of DR-TB infections in patient cohorts across five study sites in Haiti” � “In this study, we reported the occurrence of DR-TB infections in patients across five study sites in Haiti”.

- line 242 : “Secondly, we studied the phenotypic and genotypic diversity of the DR-TB strains isolated from our cohort” � “Secondly, we studied the phenotypic and genotypic diversity of the DR-TB strains isolated from patients”.

- Line 249: “In our cohort, sequencing analyses identified either T508A or silent T508T rpoB mutations in these five discrepant strain isolates, and their SIT numbers were 20 and 50 respectively, which is consistent with earlier findings(12) » � « in the tested isolates…. »

- Line 252: “In addition, strains harboring a L511P rpoB mutation have been detected in our cohort…” “In addition, strains harboring a L511P rpoB mutation have been detected, while…”

Line 239: “Thirdly, we aimed to identify the MTB clades detected in the cohort’s DR-TB patients” � “Thirdly, we aimed to identify the MTB clades of the tested DR-TB isolates”.

14. Line 269, can we make it “detection rate of HIV-TB co-infection”?

Line 286. We have modified this sentence: “Active case finding for TB and HIV should be expanded to other slum populations in Haiti as part of routine programmatic activities to increase the detection rate of TB cases” � “Active case finding for TB and HIV should be expanded to other slum populations in Haiti as part of routine programmatic activities to increase the detection rate of HIV-TB co-infection.”

15. Conclusions: Please make the conclusions in-line with the objectives.

Thank you for this suggestion, we have rephrased the Abstract conclusion (lines 46-52) and the overall article conclusion (lines 306-321) so that the link with the objectives is clearer.

16. References: Many of the references are in French.

The two references (ref. 8 and 10) are documents in French published by the Ministry of Health and Population in Haiti, they do not exist in English.

17. Supplementary material Table 1S: In the heading, Treatment column is depicted twice, please make it as Treatment after interruption and Treatment after failure.

This has been corrected.

18. Table 1S: Resistant to 3AB* - the line total should be 13.

This has been corrected and we added 3 tables in supplementary material (Table 2S, 3S et 4S).

19. Figure 1S: If you give number of isolates also, it will clarify the confusion of MDR-TB cases among total DR-TB cases.

This has been clarified by adding the following sentence: “Total of 74 DR-TB strains including 12 mono-drug resistant strains and 62 multi-drug resistant TB strains”.

20. We note that [Figure(s) 1] in your submission contain map images which may be copyrighted. 

This map has been downloaded from : https://simplemaps.com/resources/svg-ht which is a free, web-optimized, SVG Haiti map. Commercial use allowed. There is no copyright.

We have accepted and incorporated the Reviewer’s suggestions and hope that the new version of the manuscript will now be acceptable for publication in PlosOne.

Thank you very much for your kind consideration. If you have any further questions, please do not hesitate to contact me.

Sincerely,

Jonathan HOFFMANN

---

## [Decision Letter · Decision Letter 1]

4 Mar 2021

Drug-resistant TB prevalence study in 5 health institutions in Haiti

PONE-D-20-31826R1

Dear Dr. HOFFMANN,

We’re pleased to inform you that your manuscript has been judged scientifically suitable for publication and will be formally accepted for publication once it meets all outstanding technical requirements.

Kind regards,

Shampa Anupurba, MD

Academic Editor

PLOS ONE

Additional Editor Comments (optional):

Reviewers' comments:

Reviewer's Responses to Questions

**Comments to the Author**

1. If the authors have adequately addressed your comments raised in a previous round of review and you feel that this manuscript is now acceptable for publication, you may indicate that here to bypass the “Comments to the Author” section, enter your conflict of interest statement in the “Confidential to Editor” section, and submit your "Accept" recommendation.

Reviewer #1: All comments have been addressed

2. Is the manuscript technically sound, and do the data support the conclusions?

Reviewer #1: Yes

3. Has the statistical analysis been performed appropriately and rigorously? 

Reviewer #1: Yes

4. Have the authors made all data underlying the findings in their manuscript fully available?

Reviewer #1: Yes

5. Is the manuscript presented in an intelligible fashion and written in standard English?

Reviewer #1: Yes

6. Review Comments to the Author

Reviewer #1: Thank you for addressing all the comments. This time the manuscript has come up nicely and it will be useful for the readers.

7. PLOS authors have the option to publish the peer review history of their article (what does this mean?). If published, this will include your full peer review and any attached files.

Reviewer #1: **Yes: **Rahul Narang

---

## [Editor Report · Acceptance letter]

10 Mar 2021

PONE-D-20-31826R1 

Drug-resistant TB prevalence study in 5 health institutions in Haiti 

Dear Dr. Hoffmann:

I'm pleased to inform you that your manuscript has been deemed suitable for publication in PLOS ONE. Congratulations! Your manuscript is now with our production department. 

Kind regards, 

on behalf of

Dr. Shampa Anupurba 

Academic Editor

PLOS ONE